



# Quantifying fossil fuel methane emissions using observations of atmospheric ethane and an uncertain emission ratio

Alice E. Ramsden[1], Anita L. Ganesan[1], Luke M. Western[2], Matthew Rigby[2], Alistair J. Manning[3], Amy Foulds[4], James L. France[5], Patrick Barker[4], Peter Levy[6], Daniel Say[2], Adam Wisher[2], Tim Arnold[7,8], Chris Rennick[7], Kieran M. Stanley[9], Dickon Young[2], and Simon O'Doherty[2]

[1]School of Geographical Sciences, University of Bristol, Bristol, UK
[2]School of Chemistry, University of Bristol, Bristol, UK
[3]Met Office Hadley Centre, Exeter, UK
[4]School of Earth and Environmental Sciences, University of Manchester, Manchester, UK
[5]Department of Earth Sciences, Royal Holloway, University of London, Egham, UK
[6]UK Centre for Ecology and Hydrology, Edinburgh, UK
[7]National Physical Laboratory, Teddington, UK
[8]School of Geosciences, University of Edinburgh, Edinburgh, UK
[9]Institute for Atmospheric and Environmental Science, Goethe University Frankfurt, Frankfurt am Main, Germany

**Correspondence:** Alice E. Ramsden (alice.ramsden@bristol.ac.uk)

**Abstract.** We present a method for estimating fossil fuel methane emissions using observations of methane and ethane, accounting for uncertainty in their emission ratio. The ethane:methane emission ratio is incorporated as a variable parameter in a Bayesian model, with its own prior distribution and uncertainty. We find that using an emission ratio distribution mitigates bias from using a fixed, potentially incorrect emission ratio and that uncertainty in this ratio is propagated into posterior estimates of emissions. A synthetic data test is used to show the impact of assuming an incorrect ethane:methane emission ratio and demonstrate how our variable parameter model can better quantify overall uncertainty. We also use this method to estimate UK methane emissions from high-frequency observations of methane and ethane from the UK Deriving Emissions linked to Climate Change (DECC) network. Using the joint methane-ethane inverse model, we estimate annual mean UK methane emissions of approximately 0.27 (95% uncertainty interval 0.26-0.29) Tg y$^{-1}$ from fossil fuel sources and 2.06 (1.99-2.15) Tg y$^{-1}$ from non-fossil fuel sources, during the period 2015-2019. Uncertainties in UK fossil fuel emissions estimates are reduced on average by 15%, and up to 35%, when incorporating ethane into the inverse model, in comparison to results from the methane-only inversion.

## 1 Introduction

Atmospheric methane (CH$_4$) is a potent greenhouse gas with many natural and anthropogenic sources. These sources can be split into three main types: microbial methane which is emitted during the decomposition of organic matter, pyrogenic methane which is formed during incomplete combustion of biomass, and thermogenic methane which is released from fossil fuels during their extraction, refinement and use. Globally, anthropogenic sources account for approximately 60% of total methane





emissions. The largest sources of methane are agriculture and waste management (approximately 35% of total emissions) and fossil fuel production and use (approximately 20% of total emissions) (Saunois et al., 2020).

Methane has contributed to approximately 25% of the total anthropogenic radiative forcing caused by warming agents since pre-industrial times (Myhre et al., 2013). Due to its short atmospheric lifetime and high impact on radiative forcing in the atmosphere, reduction in methane emissions is a key target for many countries (Ganesan et al., 2019).

Despite its importance when considering climate change targets, concentrations of methane in the atmosphere are continuing to rise rapidly. Recent years have seen an acceleration in this upwards trend, with a global annual increase in atmospheric

methane concentration of approximately 15 parts per billion (ppb) between 2019 and 2020 (Dlugokencky and NOAA/GML, 2021). There is no established consensus over the cause of the recent increase in atmospheric concentration, with studies suggesting increases in tropical wetland emissions (Bousquet et al., 2011; Nisbet et al., 2019; Schaefer et al., 2016), potential changes to the hydroxyl radical concentration (Rigby et al., 2017; Turner et al., 2017; Thompson et al., 2018) and variation in fossil fuel emissions (McNorton et al., 2018; Thompson et al., 2018; Hausmann et al., 2016) as possible contributors. The

variety of proposed mechanisms for recent changes in atmospheric methane highlights why this is a key area for research.

In this work, we present a method to quantify methane emissions with improved uncertainty characterisation through inverse modelling of atmospheric observations. Estimates of sector-level emissions are calculated using observations of a secondary trace gas and its emission ratio relative to methane. The key development in this work over previous methods is the inclusion of an emission ratio as a variable parameter, which is inferred along with sectoral methane emissions using a Bayesian inversion

framework.

A general discussion of methane emissions estimation and a review of previous work using ethane for fossil fuel emissions estimation is provided in the rest of Sect. 1. We discuss our statistical method in Sect. 2. The methods used for a synthetic data experiment and for a case study on UK's fossil fuel methane emissions are described in Sect. 2.1 and 2.2. Results from these two experiments are given in Sect. 3 and discussions of the results in Sect. 4, followed by our concluding remarks.

**1.1 Estimating methane emissions**

Methane emissions can be estimated using two main approaches: 'bottom-up' and 'top-down' modelling. Bottom-up methods model the physical and chemical processes of methane emission to create estimates of sector-level emissions, which can be distributed in space and time at a range of frequencies. However, methane emissions inventories have been shown to be inaccurate in some cases when compared to observations, which could lead to an incorrect representation of methane sources.

For example, the spatial distribution of methane emissions from oil and gas sources in the Emissions Database for Global Atmospheric Research (EDGAR) (Team EDGAR, 2021) were shown to be too heavily weighted towards locations where these fuels were distributed and used, rather than areas of fossil fuel extraction and production (Chen et al., 2018). In the United States, inventory data from EDGAR and the US Environmental Protection Agency (EPA) have repeatedly been found to underestimate methane emissions from both fossil fuel and agricultural sectors, when compared to flux estimates derived

from observations, (Miller et al., 2013; Alvarez et al., 2018; Barkley et al., 2019). With a rapidly changing oil and natural gas industry worldwide (e.g U.S. Energy Information Administration, 2021), and the long time periods required to produce





and verify bottom-up inventories, real-time assessment of fossil fuel methane emissions using bottom-up inventories can be challenging.

Top-down estimation of emissions uses observations of methane concentrations in the atmosphere and a chemical transport model to infer fluxes, often through Bayesian methods. These observations can be directly sampled from ambient air or remotely sensed. An estimate of emissions from a 'bottom-up' model is typically used as prior information to inform the top-down inverse model during the inference of a posterior emissions distribution, and to partition emissions to their source based on their location.

Measurements of additional trace gases can be used with a top-down approach to partition emissions, when these tracer gases are co-emitted with methane from a particular source at a characteristic ratio. For example, carbon monoxide (CO) is co-emitted with methane during incomplete combustion (Heald et al., 2004) so could be used to quantify emissions from biomass burning. Ethane ($C_2H_6$) is emitted by fossil fuel production and use and has no significant emissions from biogenic sources (Peischl et al., 2013; Helmig et al., 2016), so can be used to quantify fossil fuel methane emissions. Methane isotopologue observations (e.g. $^{13}CH_4$) can be utilised to apportion emissions in a similar method, by considering the ratio of isotopologues emitted from each source type (Milkov et al., 2020; Lan et al., 2021). Studies have shown that when incorporating emission ratios or observations of additional gases into emissions quantification frameworks, the uncertainty in emissions estimates of the primary gas can be reduced significantly when compared to a single gas model (Palmer et al., 2006; Wang et al., 2009; Boschetti et al., 2018). However, these approaches always require a thorough understanding of the associated emission ratios, as inaccuracies in these values could introduce large posterior errors (as discussed in Nathan et al., 2018) or lead to emissions being incorrectly partitioned (Schwietzke et al., 2016; Sherwood et al., 2017).

## 1.2 Previous work using ethane observations to infer fossil fuel methane emissions

Previous studies have used ethane observations and emission ratios in a range of methods for the source partitioning of methane emissions. Typically, the enhancement in aircraft mole fraction observations of methane and ethane are compared to a bottom-up estimate of an ethane:methane ratio to assign a proportion of total regional methane emissions to a fossil fuel source (e.g. Baier et al., 2020; Mielke-Maday et al., 2019). Similar methods have also been used more locally over cities or individual gas fields, where comparisons between literature and observed emission ratios have been used for source attribution of methane enhancements seen in individual aircraft-observed plumes (Yacovitch et al., 2017; Lowry et al., 2020).

Ethane observations have also been incorporated more directly into joint inverse models, where emissions are optimised simultaneously to create emissions profiles characteristic of each source type (Peischl et al., 2013; Kuwayama et al., 2019). Ethane and methane aircraft observations have also been optimised in a joint model to estimate surface methane fluxes, by comparing the observed ethane:methane emission ratios to a bottom-up estimate of the ratio (Barkley et al., 2019).

Most of this previous work using observations of methane and ethane has used a fixed estimate of the ethane:methane emission ratio as a basis for the apportionment of methane emissions. Whilst some have considered trends in the ratio (Wunch et al., 2016), most studies assume that this ratio is constant, which is unlikely to be true in most situations (Hausmann et al., 2016; Lan et al., 2019; Nisbet et al., 2019) as the ratio can vary with location and over time, depending on the type of fossil





fuel source and the type of extraction or processing techniques being used. Incorrectly assuming that this ratio is fixed could introduce errors into any sector-level emissions estimates, and could alter the inference of emission trends.

## 2 Methods

In this work, a top-down hierarchical Bayesian inverse model uses observations of a secondary trace gas and its emission ratio
with respect to a primary gas, to solve for emissions of the primary gas at a sectoral level. Uncertainties in the emission ratio between the primary and secondary gases are statistically propagated into the emissions distributions through the hierarchical framework. The principle of this method is described below.

A forward model (Eq. 1) links observed mole fractions of a gas $\mathbf{y}$ to its emissions, $\mathbf{x}$ via a linear atmospheric chemistry and transport model $\mathbf{H}$ and model-measurement error $\epsilon$. $\mathbf{x}$ is inferred through an 'inversion' of the forward model using Bayesian
statistics.

$$\mathbf{y} = \mathbf{Hx} + \epsilon. \tag{1}$$

Prior probability density functions (PDFs) must first be assigned to the parameters. To reduce the subjectivity involved when choosing these PDFs, additional 'hyper-parameters' can be included in a hierarchical Bayesian framework, which place distributions on these uncertain parameters rather than imposing them as fixed values. Ganesan et al. (2014) found that by including
uncertainty in parameters (such as model-measurement error) as hyper-parameters, one could better propagate uncertainties into the posterior estimate of emissions. To use these hyper-parameters in the inverse model, Bayes's theorem is extended to include the joint distributions between primary and secondary parameters $\theta$ (Ganesan et al., 2014),

$$\rho(\mathbf{x}|\mathbf{y}) \propto \rho(\mathbf{y}|\mathbf{x},\boldsymbol{\theta}) \cdot \rho(\mathbf{x}|\boldsymbol{\theta}) \cdot \rho(\boldsymbol{\theta}). \tag{2}$$

There is no analytical solution to maximise Eq. 2 so a Markov Chain Monte Carlo (MCMC) method is used produce
a posterior distribution containing possible solutions for each of the parameters. This is an iterative method that randomly samples the PDFs of the parameters involved, then accepts or rejects these new parameter values based on their probability density, relative to the prior and observation distributions (Ganesan et al., 2014). The step sizes used to dictate the size of the sampling distribution for each parameter are optimised through an adaptive MCMC process to produce an acceptance ratio of approximately 0.35, using an adapted version of Algorithm 4 from Andrieu and Thoms (2008). The first 50% of these samples
are discarded as a 'burn in' period to remove memory of the initial state and every $100^{th}$ value of the remaining samples is retained to form posterior distributions for the optimised parameters. With MCMC methods, non-Gaussian distributions can be used to represent the input parameters; for example, a less well understood parameter may be better represented by a uniform distribution, where upper and lower bounds of the distribution can be set to cement the solution in physical terms.





To solve for emissions from separate sources, the forward model is expanded to include emissions of the primary gas from
two sectors *A* and *B*:

$$\mathbf{y}_{\text{Gas1}} = \mathbf{H}_{\text{Gas1},A} \cdot \mathbf{x}_{\text{Gas1},A} + \mathbf{H}_{\text{Gas1},B} \cdot \mathbf{x}_{\text{Gas1},B} + \boldsymbol{\epsilon}_{\text{Gas1}}. \tag{3}$$

Observations of the secondary gas and its emission ratio are incorporated into this model as follows. Assuming that Gas 2 is
only co-emitted from sector *A*, with an emission ratio $\mathbf{R}$ relative to Gas 1, the forward model for Gas 2 is expressed as:

$$\mathbf{y}_{\text{Gas2}} = \mathbf{H}_{\text{Gas2},A} \cdot \mathbf{R} \cdot \mathbf{x}_{\text{Gas1},A} + \boldsymbol{\epsilon}_{\text{Gas2}}. \tag{4}$$

In an application where a particle dispersion model is used to provide transport model 'footprints' (as is the case for the
remainder of this work) and when analysing observations of gases with long atmospheric lifetimes, atmospheric transport of
both gases can be assumed to be equivalent. Therefore, the linear transport model is the same for both gases and is represented
from this point onward as $\mathbf{H}$.

Combining the two forward models, Eq. 3 and 4, produces a joint model where both gases inform the estimate of emissions:

$$\begin{bmatrix} \mathbf{y}_{Gas1} \\ \mathbf{y}_{Gas2} \end{bmatrix} = \begin{bmatrix} \mathbf{H}_A & \mathbf{H}_B \\ \mathbf{R} \cdot \mathbf{H}_A & \mathbf{0} \end{bmatrix} \begin{bmatrix} \mathbf{x}_{\text{Gas1},A} \\ \mathbf{x}_{\text{Gas1},B} \end{bmatrix} + \begin{bmatrix} \boldsymbol{\epsilon}_{Gas1} \\ \boldsymbol{\epsilon}_{Gas2} \end{bmatrix}. \tag{5}$$

Without a framework that can consider the uncertainty in the emission ratio, $\mathbf{R}$ would be imposed as a fixed parameter
into the sensitivity matrix at this point. In our work, the emission ratio $\mathbf{R}$ is treated as a variable parameter, requiring the
expansion of Bayes's theorem as discussed above and shown in Eq. 6. Model-measurement uncertainty ($\boldsymbol{\sigma}_y$) is also included
as a hyper-parameter, again with its own prior PDF and uncertainty,

$$\rho(\mathbf{x}, \mathbf{R}, \sigma_y | \mathbf{y}) \propto \rho(\mathbf{y} | \mathbf{x}, \mathbf{R}, \boldsymbol{\sigma}_y) \cdot \rho(\mathbf{x}) \cdot \rho(\mathbf{R}) \cdot \rho(\boldsymbol{\sigma}_y). \tag{6}$$

A MCMC process is used to produce posterior distributions for both the emissions and emission ratio parameters. In this
study, we used this model to estimate methane emissions from fossil fuel (*FF*) and non-fossil fuel (*non-FF*) sources, using
ethane as the secondary gas. However, this model framework is highly adaptable and could be used with other tracers, for
example, methane isotopologues.

## 2.1 Synthetic data experiment

To investigate the influence of the ethane:methane emission ratio on posterior estimates of methane emissions, we carried out
model runs as described above, using synthetic data generated from a known emissions field and a known emission ratio.
These tests used a two-sector model of identical UK 'fossil fuel' (*FF*) and 'non-fossil fuel' (*non-FF*) fluxes, with the same





magnitude and spatial distribution of emissions. Total UK methane emissions from the UK National Atmospheric Emissions

Inventory (NAEI) (https://naei.beis.gov.uk/) were used to represent emissions from both sectors. This test simulates a scenario when fluxes from both sectors are inseparable by spatial differences alone. For these synthetic data tests we did not consider background levels of methane (i.e., the contribution to the total mole fraction from emissions outside the UK) and only tested the ability of the inversion to return the regional (UK) emissions field.

The a priori ethane:methane emission ratio, **R**, was assumed to be uniform across the whole domain, with a value of 0.075.

This is the approximate mean ethane:methane emission ratio from natural gas sources in Europe, (Table 1). We assumed no ethane emissions from the *non-FF* sector.

We created four-hourly synthetic methane observations at four UK tall tower sites and one coastal site in the UK Deriving Emissions linked to Climate Change (DECC) network (Stanley et al., 2018; Stavert et al., 2019) at Mace Head (MHD), Tacolneston (TAC), Bilsdale (BSD), Ridge Hill (RGL) and Heathfield (HFD) (see Appendix Fig. A1 for locations) by com-

bining the synthetic emissions fields with atmospheric transport footprints made using the Met Office's Lagrangian Numerical Atmospheric-dispersion Modelling Environment (NAME) (Jones et al., 2007). See Appendix A for details on how NAME was run and for an example transport footprint. Synthetic ethane observations were created by combining *FF* ethane emissions (generated as *FF* methane emissions times the known uniform emission ratio of 0.075) with the transport model footprints. To mirror the DECC network, methane observations were created for all five sites but ethane observations were only created for

two sites, MHD and TAC. For both gases, Gaussian noise with a standard deviation equal to 10% of each measurement was added to simulate instrument noise and model error.

Three sets of inversions were run:

1. Joint methane-ethane inversions where the emission ratio was fixed at values ranging from 0.5-1.5 times the true value. This test simulates studies that hard-wire emission ratios at potentially incorrect values, without considering their uncer-

tainty.

2. Joint methane-ethane inversions where the emission ratio is a variable parameter with its own PDF representing the range of uncertainty in the emission ratio. The emission ratio prior PDF was given a uniform distribution ranging from 0.5-1.5 times the true value. This simulates the situation where uncertainty in the emission ratio is built into the framework.

3. A methane observation only (i.e. one gas) inversion where no ethane observations or emission ratio are included and the

attribution of emissions is only informed by the spatial distinction of sources in the prior (which in this case, does not exist).

In all tests, the inversion solved for emissions as a scaling of the a priori emissions field, using basis functions to coarsen the native grid resolution of the transport model onto a 7×7 grid over the UK, with the rest of the European domain split into four larger regions. Ethane:methane emission ratios were solved for at the same resolution as methane emissions. Gaussian

distributions were used for emissions parameters in these synthetic data tests. As the true emissions field is known here and to represent a real-world situation where the prior mean may not necessarily be the true value, we used emission PDFs with



| Molar ratio (median and range) | Source type | Reference |
|---|---|---|
| 0.045 (0-2.76) | Global conventional oil and gas composition | (Sherwood et al., 2017) |
| 0.038 (0.001-1.0) | European raw gas composition | (Visschedijk et al., 2018) |
| 0.03 | UK gas and oil distribution | (Xiao et al., 2008) |
| (0.049-0.09) | UK gas leaks | (Lowry et al., 2020) |

**Table 1.** Estimates of ethane:methane molar emission ratios from a range of fossil fuel methane sources.

a priori means equal to 125% and 75% of their true values for the *FF* and *non-FF* sectors, respectively, to simulate slightly incorrect a priori emissions fields (i.e. correct total emissions but incorrect partitioning). Both sectors were given a standard deviation of 50% of their true values. Model-measurement uncertainty was fixed at 10% of the mean pseudo-observation value

for both methane and ethane.

## 2.2   UK methane emissions case study

We used the methane-only and joint methane-ethane inverse models to estimate monthly UK methane emissions from 2015 to 2019. We also tested the impacts of a fixed emission ratio on posterior flux estimates and investigated the propagation of uncertainties through the model when applying an uncertainty to this emission ratio.

### 2.2.1   Observations and transport footprints

Methane observations were used from the five current UK DECC network sites as discussed in Sect. 2.1. Mole fraction observations of methane were made using Cavity Ringdown Spectroscopy (CRDS) instruments Picarro G2301 and G2401, calibrated using daily standard measurements, and are reported on the WMO-X2004A scale (Stanley et al., 2018; Stavert et al., 2019). Ethane observations were made at two DECC sites, MHD and TAC, using a Medusa gas chromatography-mass spectrome-

try (GCMS) instrument (Prinn et al., 2018). Calibration of ethane observations is currently based on the provisional SIO-p (Scripps Institution of Oceanography) scale. Frequent comparisons between Advanced Global Atmospheric Gases Experiment (AGAGE) ethane measurements (for example those made at the MHD site) and those reported by the National Oceanic and Atmospheric Administration (NOAA) at the same site, but using an independent calibration scale, show no significant long-term bias. A complete description of the ethane calibration employed here is given in Mühle et al. (2007).

Observations from the highest inlet at each tall tower site were used to reduce the impact of local fluxes and to increase the size of the footprint, with the exception of 2015-2016 for ethane, when the instrument measured from the middle inlet (100 magl) at TAC. A complete discussion of instrumentation, inlet heights and uncertainty characterisation is presented in Stanley et al. (2018) for the MHD, TAC and RGL sites, and in Stavert et al. (2019) for HFD and BSD.

Observations of both gases were filtered to remove points when local emissions are likely to bias results, using similar

methods to as described in Lunt et al. (2021). Measurements made at times when the tower inlet was sampling air from above the planetary boundary layer were removed. Measurements were also removed when more than 10% of the area-integrated





sensitivity at the site was from the 25 grid cells surrounding the site (i.e. local sources). Remaining observations were averaged into four-hourly periods. On average 40 (range 18 - 69) % of observations were filtered each month.

The NAME model was used to produce transport 'footprints' for all observation sites. See Appendix A for more detail on
how NAME was run and for an example footprint for the network of sites. As methane's lifetime of around a decade is long
compared to the timescale of transport within the regional domain (on the order of days), we assumed that atmospheric loss
is negligible and that only transport influences the relationship between surface emissions and atmospheric concentrations.
Ethane has a shorter lifetime than methane (from approximately 2 months in summer to 6 months in winter (Helmig et al.,
2016)). However, we found atmospheric loss of ethane on a 2-month timescale to have a negligible effect on the footprints over
the UK and therefore we used the same transport footprints for both gases.

### 2.2.2   Model parameters and a priori PDFs

The a priori estimate of UK methane emissions from each sector was taken from the UK Greenhouse Gas (UKGHG, Levy,
2020) model of spatially and temporally disaggregated emissions, which is based on national, annual totals from the UK
National Atmospheric Emissions Inventory (NAEI). Emissions from the sectors 'energy production', 'offshore', 'industrial
and domestic combustion', 'industrial processes', 'road transport' and 'other transport' were summed to form an a priori
field for *FF* emissions. Emissions from 'agriculture', 'waste' and 'natural' sectors formed the a priori field for the non-FF
sector. Emissions from areas outside the UK but within the modelling domain, including for example Western Europe, were
taken from the Emissions Database for Global Atmospheric Research (EDGAR) v5.0 (Crippa et al., 2020; Team EDGAR,
2021). The spatial distribution and percentage contribution from each source to total emissions from each grid cell are given in
Appendix Fig. C1.

Boundary condition 'curtains' representing the methane mole fractions at the edges of the study domain, were derived from
the global methane model CAMS v19r1 (available via https://ads.atmosphere.copernicus.eu/). Spatially uniform boundary
condition curtains were used for ethane, based on the monthly mean ethane concentration observed at MHD.

Scaling factors to the a priori emissions and emission ratios were solved for at a coarser resolution than the resolution of the
transport model. The study domain was split into 49 regions using a quadtree algorithm (see e.g. Western et al., 2021) which
placed a higher density of smaller grid cells in areas with greater sensitivity to emissions and a lower density of larger cells
in areas with less sensitivity to emissions. The inversion then solved for a scaling factor of the a priori emissions from each
sector and an emission ratio for each of the 49 regions, for each calendar month. Four boundary condition scaling parame-
ters representing adjustments to the curtains at each horizontal boundary were also estimated for each gas, also at monthly
resolution.

Prior distributions for emissions scaling factors were assumed to be Gaussian, truncated at zero to prevent the model from
converging on negative emissions. Prior emissions scaling factor PDFs were given a mean of 1 and standard deviation of
0.5 (before truncation). Boundary condition scaling factors for the four horizontal boundaries for each gas were also given
Gaussian PDFs, truncated at zero, with a mean of 1 and an uncertainty of 0.05 and 0.5 (before truncation) for methane and





ethane, respectively. Ethane boundary condition uncertainties are assumed to be large due to their large seasonal and latitudinal

variations.

Uncertainty in the observations were calculated as the quadratic sum of measurement uncertainty and model uncertainty. Measurement representation uncertainty was taken to be the variability of 1 minute of data, within the 4 hour averaging period. Model uncertainty was included as a hyper-parameter, with one value per site per month solved for during the inversion. This

model-measurement uncertainty was given a uniform PDF, between 10 and 50 ppb for methane and between 20 and 50 ppb for ethane. A full description of a similar use of model uncertainty in a hierarchical framework can be found in Ganesan et al. (2014).

A uniform a priori emission ratio PDF was used for each of the 49 regions with bounds of 0.0075 and 0.2. These values were chosen to include the most common ratios found by bottom up estimates of European fossil fuel ethane:methane ratios from a

range of studies and databases (Table 1).

## 3   Results

### 3.1   Synthetic data experiment results

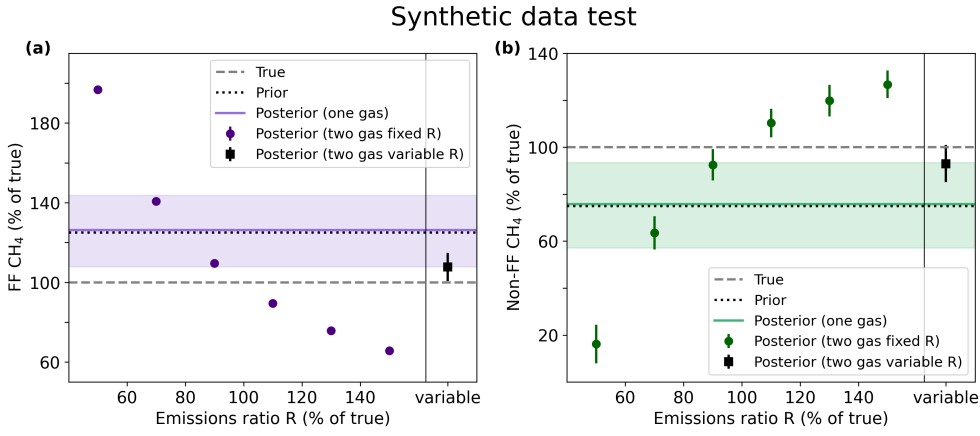

**Figure 1.** Posterior methane emissions expressed as means and 95% confidence intervals from the synthetic data tests. Methane-only model (solid line and shading), joint methane-ethane model (dots and error bars) with a range of fixed emission ratios and a variable emission ratio (furthest right point in both (a) and (b)). *FF* fluxes on the left (a) in purple and *non-FF* fluxes on the right (b) in green. The true and a priori mean fluxes are given as dashed and dotted grey lines, respectively.

Results from synthetic data tests, showing the impacts of a fixed and variable emission ratio, are summarised in Fig. 1. Because there is no spatial distinction between sources in the prior and because the total posterior emissions are the true total,

the methane-only (one gas) model returns the prior mean emissions for each sector. This lack of sectoral information from the prior is also expressed in the relatively large posterior uncertainties for both sectors.





In the joint methane-ethane inversion, there is more information available for the model to constrain emissions from each source. However, when the emission ratio **R** is fixed in the inversion at an incorrect value, the sectoral partitioning of emissions is also incorrect but is derived with high confidence. If the emission ratio is fixed at a value 50% lower than its 'true' value,

posterior mean *FF* fluxes are estimated to be over 80% larger than their true value. As total emissions are constrained by the methane observations, the estimate for *non-FF* fluxes is therefore skewed in the opposite direction, with posterior mean fluxes smaller than their true value. 95% confidence intervals on *FF* emissions in this test are too small to be visible on this scale, due to the high level of constraint from the fixed emission ratio. This synthetic data test highlights how errors could be introduced when using a fixed ethane:methane ratio that does not reflect the true uncertainty in the parameter.

Results from the joint methane-ethane model that considers the uncertainty in the emission ratio (Fig. 1 furthest right points in both (a) and (b)) show that the potential errors introduced by assuming an incorrect ratio can be mitigated by including **R** as a variable parameter. In this case, posterior fluxes from both sectors converge closer to the true sector-level emissions, with a reduced posterior uncertainty compared to the methane-only model output but with larger uncertainty than if fixing the emission ratio. True emissions are not replicated exactly as there is some small dependence on the emissions prior. *FF* emissions are

constrained by both methane and ethane observations, so most of the uncertainty in **R** is therefore carried forward into the estimates of *non-FF* fluxes.

### 3.2 UK monthly methane emissions 2015 - 2019

We used the joint methane-ethane inverse model to create posterior estimates of the UK's monthly *FF* and *non-FF* methane emissions for 2015 to 2019. The methane-only model was run for the same period for comparison. Figure 2 gives the monthly

posterior flux from the UK for the *FF* and *non-FF* sectors. Because total methane emissions are constrained by the methane observations, posterior total emissions from the methane-only and joint methane-ethane inversions are equal. The differences in results are shown in the partitioning of emissions from the two sectors.

The joint methane-ethane inversion finds that emissions from *FF* sources contribute on average 15% less to total methane emissions than in the methane-only inversion. This is balanced by a proportional increase in *non-FF* emissions. The impact

on posterior uncertainty varies across the period, with an average 15% reduction in the size of the posterior *FF* flux 95% uncertainty interval, which increases up to 35% for some months. Our results show declining emissions over the time period, which is largely driven by emissions from *non-FF* sources. Annual mean posterior flux estimates are given in Table 2. These results are consistent with total emissions derived in previous inverse modelling studies using the same data (Western et al., 2020; Lunt et al., 2021).

A comparison between observed methane and ethane mole fractions and posterior modelled mole fractions from the joint methane-ethane model for two example months (April-May 2019) is given in Fig. 3. Percentage differences between observed and modelled mole fractions across the time-series are given as histograms for each site. Baseline mole fractions from all methane sites (dashed lines in Fig. 3) are consistent with those from the background site (MHD). Appendix Fig. D1 shows a scatter plot comparing observed and modelled posterior mole fractions from the joint methane-ethane model for the full time-

series from 2015-2019. There is generally a good fit to observations, but the model does not always fit to the largest methane

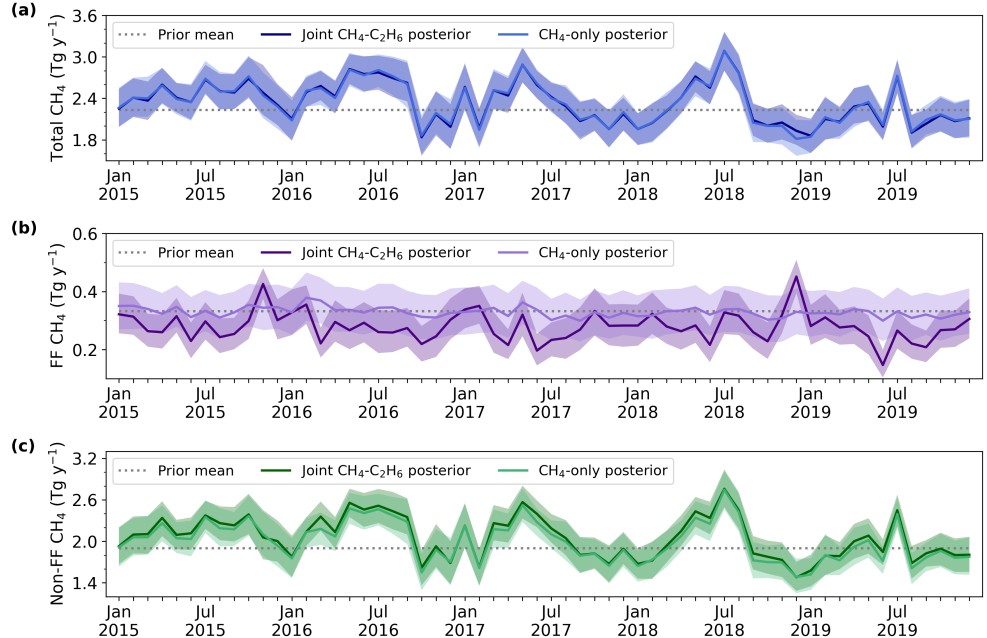

**Figure 2.** Posterior monthly UK methane emissions in Tg y$^{-1}$. Total (a, blue), FF (b, purple) and non-FF (c, green), expressed as posterior means and 95% uncertainty intervals of these PDFs. Methane-only model output (lighter shade line and shading) and joint methane-ethane model output (darker shade line and shading) both shown for comparison. A priori mean fluxes from the UKGHG model are given as a grey dashed line.

| Year | FF CH$_4$ (Tg y$^{-1}$) | Non-FF CH$_4$ (Tg y$^{-1}$) | Total CH$_4$ (Tg y$^{-1}$) |
|------|------------------------|-----------------------------|----------------------------|
| 2015 | 0.28 (0.26-0.31) | 2.16 (2.00-2.30) | 2.44 (2.32-2.56) |
| 2016 | 0.28 (0.24-0.30) | 2.18 (1.98-2.37) | 2.46 (2.28-2.64) |
| 2017 | 0.26 (0.24-0.28) | 2.05 (1.88-2.21) | 2.31 (2.16-2.47) |
| 2018 | 0.29 (0.26-0.32) | 2.01 (1.82-2.21) | 2.29 (2.11-2.48) |
| 2019 | 0.25 (0.23-0.28) | 1.90 (1.78-2.04) | 2.15 (2.03-2.28) |

**Table 2.** Results from the joint methane-ethane inversion. Annual posterior UK methane emissions from FF and non-FF sectors, given as posterior means and 95% uncertainty intervals.

peaks from TAC, RGL and HFD, the three sites closest to areas of high emissions. Comparisons between observations and an a priori estimate of mole fractions made by combining the a priori map of fluxes with the transport model are also given in Appendix Fig. D1. Overall, there is an improved fit to both methane and ethane observations in the posterior estimate of mole fractions produced by the inverse model.





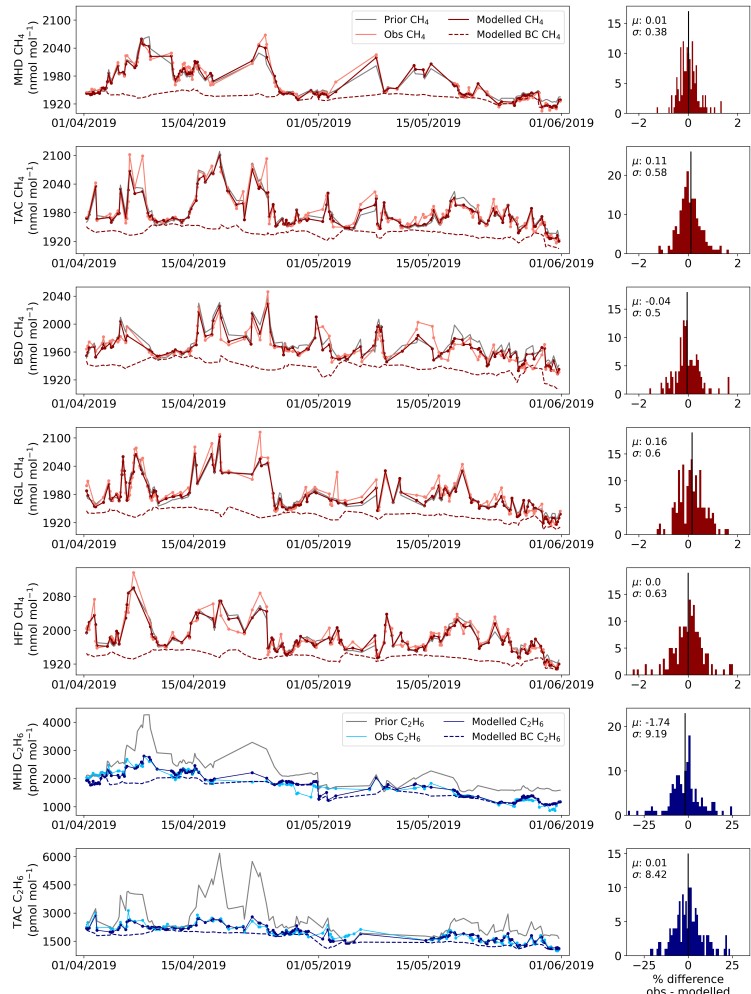

**Figure 3.** Observations (dots) and modelled observations (solid lines) of methane (red) and ethane (blue) from the UK DECC network for April-May 2019. Modelled boundary condition (baseline) emissions are also given (dashed line) along with the a priori modelled concentrations (grey). Note the different scales for each site and for each gas. A histogram showing the percentage differences between observed and modelled mole fractions is provided for each gas and site.

Model uncertainty for methane mole fractions converged at similar values for all sites across the time period, with a mean value of 7.75 ppb overall (with 75% of all mean model error values between 5 and 10 ppb). Due to the high peaks and troughs in ethane observations, the model consistently attempted to converge ethane model measurement uncertainty at the upper bound of its prior uncertainty range.

The a priori emissions and average spatial distribution of posterior emissions scaling factors for 2019 are shown in Fig. 4.
The ethane observations only indirectly constrain the much larger *non-FF* emissions, so there little difference in the spatial



distribution of *non-FF* emissions between the methane-only and joint methane-ethane inversions. However, the joint methane-ethane inversion suggests a different distribution of *FF* emissions, where emissions are scaled down in most locations, apart from a few regions with large positive scaling which often correlate with heavily populated areas (e.g. the West Midlands, London and south coast of England).

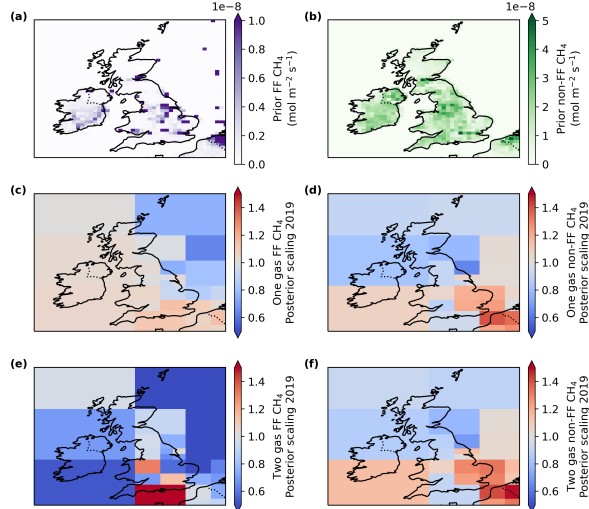

**Figure 4.** A priori emissions and 2019 annual mean posterior emissions scaling factors for UK *FF* (left) and *non-FF* (right) methane fluxes. A priori fluxes from the UKGHG model, as described in the text, are given at the resolution of the transport model (a,b). Posterior mean flux scaling factors from the methane-only (c,d) and joint ethane-methane (e,f) inverse models are given at the coarser resolution of the transport model. Red and blue indicate a scaling up or down, respectively, of the a priori estimate.

Across the whole period of study, posterior mean ethane:methane emission ratios varied across the domain between 0.009 and 0.2. For example, in July 2019, approximately 20% of **R** values converged with clear Gaussian posterior distributions, suggesting a strongly correlated relationship between the two gases that the model was able to use to inform the posterior distribution of the both the methane flux and ethane:methane ratio in that area. Posterior emission ratio PDFs in the remaining areas of the domain were more uniform, indicating a weak constraint from the observations in those areas. Mean posterior

emission ratios for two different periods are shown in Fig. 5. Some regions, for example, central south England, London and the West Midlands, often converge with high emission ratios close to the upper bound of the a priori PDF. Basis functions where the posterior uncertainty in **R** (defined as posterior $95^{th}$ percentile divided by posterior mean) is less than 50% of the prior uncertainty (prior $95^{th}$ percentile divided by prior mean) are highlighted in Fig. 5, showing areas where the observations were most able to constrain **R**.



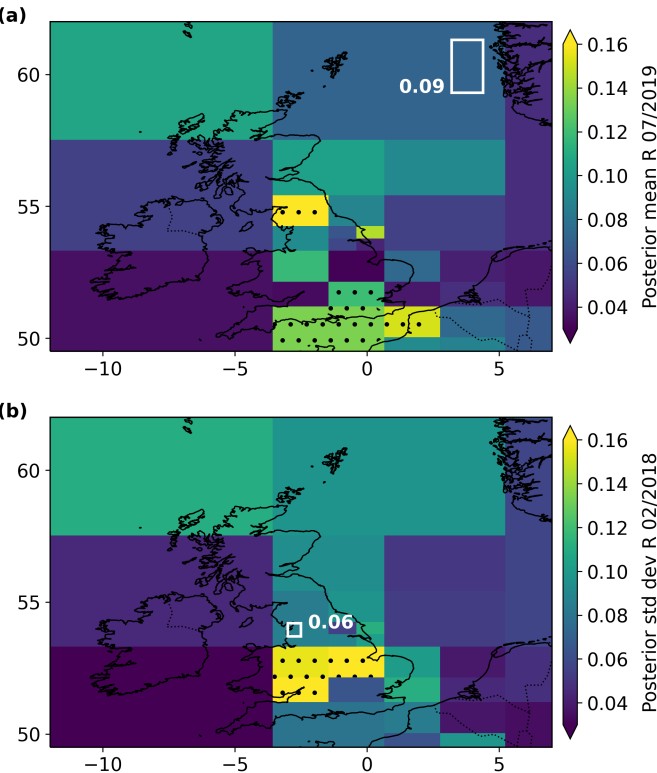

**Figure 5.** Comparison of posterior emission ratios **R** from the joint methane-ethane inverse model with independently observed ethane:methane ratios labeled in white boxes. Average emission ratio for July 2019 and comparison with ratios derived from plumes observed during FAAM flight C191 (a). Average emission ratio for February 2018 and comparison to observations from Royal Holloway Mobile Laboratory sampling (b). Basis functions where the posterior uncertainty (defined as posterior $95^{th}$ percentile divided by posterior mean) is less than 50% of the prior uncertainty (prior $95^{th}$ percentile divided by prior mean) are filled with a hatching of black dots.

### 3.3 Comparison of posterior emission ratios with independent measurements

We compare posterior ethane:methane emission ratios from the hierarchical inverse model to independent calculations of this ratio, made during a range of mobile observation studies. These independent datasets are sparse and thus only a limited validation can be performed.

Ethane and methane airborne observations were made during Flight C191 of the Facility for Airborne Atmospheric Measurements (FAAM) campaign over North Sea oil and gas fields on 29 July 2019, as part of the Methane Observations and Yearly Assessments (MOYA) project. A mean average ethane:methane emission ratio of 0.088 (range 0.04-0.18) was calculated from 4 different plume observations using two different methods, Gaussian plume fitting and linear regression. See Appendix B for information on how these ratios were calculated. In addition, the Royal Holloway mobile laboratory sampled air around potential shale gas production sites for baseline monitoring (Lowry et al., 2020). They observed ethane:methane ratios of





approximately 0.06 from local gas leaks, on 27 February 2018. Comparisons between these individual plume estimates and our monthly posterior mean emission ratios are shown in Fig. 5 (a,b). Both independently measured ratios are approximately consistent with the emission ratios estimated in this work. However, observations from Flight C191 are located far from the DECC observation network so our estimates of emission ratios over the North Sea have likely to have larger uncertainties than those closer to the towers.

As most independent observations of ethane:methane ratios over the UK have only been taken over short time periods, this limits the scope of comparison available with our monthly model estimates of these ratios. Partitioning of the domain into coarse basis functions could also impact the comparison, as ratios are likely to be heterogeneous within a basis function region. As we are focused on average emissions over the month, this should not significantly affect our results, but could limit the ability for further validation of our posterior emission ratios.

### 3.4 Impact of a fixed ethane:methane emission ratio on UK methane fluxes

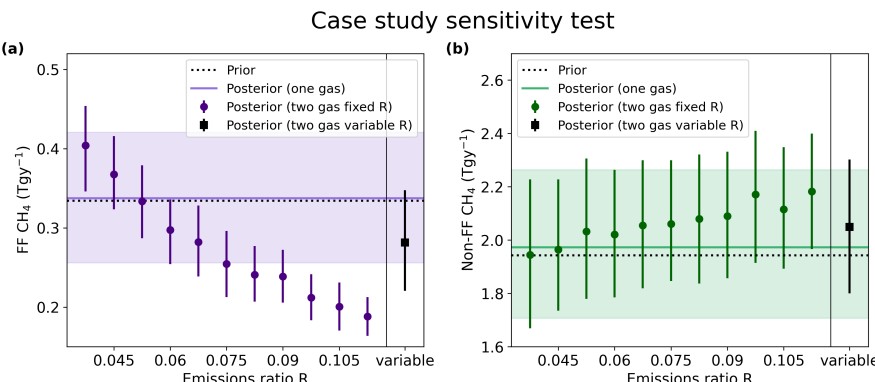

**Figure 6.** Posterior UK methane fluxes for May 2015, expressed as means and 95% uncertainty intervals. Methane-only model (solid line and shading), joint methane-ethane model (dots and error bars) with either a fixed emission ratio or a variable emission ratio (farthest right value in both (a) and (b)). *FF* fluxes on the left (a) in purple and *non-FF* fluxes on the right (b) in green. The a priori mean fluxes are given as dashed grey lines.

As in the synthetic data test, we tested the impact of using a fixed ethane:methane ratio on one month of posterior UK sectoral methane fluxes (Fig. 6). We ran the model for one month (April 2019), but used a range of spatially uniform emission ratios (**R**). As in the synthetic data test results, posterior fluxes are strongly influenced by a fixed emission ratio. For example, by assuming a fixed ratio scaling factor of 0.5 (which equates to an emission ratio of approximately 0.04, similar to literature

values for natural gas fossil fuel methane sources) the estimate of mean posterior *FF* flux is approximately 60% higher than when using a fixed ratio of 0.075 (approximately the mean emission ratio from a range of studies e.g. Table 1). As the rightmost points in both Fig. 6 (a,b) show, the joint methane-ethane inversion with a variable emission ratio samples the uncertainty in the





emission ratio and propagates this into the posterior flux estimates. Uncertainties in the posterior flux estimates are therefore higher for both sectors than when using a fixed emission ratio, but capture the overall uncertainty in the system more accurately.

**4    Discussion**

This work demonstrates the potential advancements that can be made in sector-level emissions estimation when incorporating observations of a secondary co-emitted tracer into an inverse model, but only when considering the uncertain nature of emission ratios. Both our synthetic data and UK tests show that over-confidence in knowledge of emission ratios can bias the model toward incorrect source partitioning. We also show that in the UK, the joint methane-ethane model suggests a different spatial

distribution of *FF* and *non-FF* emissions than reflected in the a priori estimate, which would be the sole constraint on sector partitioning in a methane-only inversion.

One limitation of this study is that we assume that there are no ethane emissions from sources other than those co-emitted with methane from the *FF* sector. Small amounts of ethane are emitted naturally from geological seeps and during biomass burning (Nicewonger et al., 2016) but this should have negligible contribution to ethane emissions over our study domain. We

also assume a single emission ratio for all fuel types. A more detailed partitioning of methane sources into sub-sectors of fossil fuel emissions may also be possible with the model, if more specific emission ratios are considered and with a higher density of ethane observations. Our emission ratios are estimated monthly which does not account for any short-term changes in ratios seen from, for example, flaring or gas leaks.

This study has highlighted the effectiveness of the high density observational network in and around the UK for estimating

regional methane emissions. The methane-only model was able to produce total methane emissions estimates consistent with previous top-down estimates of UK emissions from similar years (Ganesan et al., 2015; Western et al., 2020). The difference between methane-only and methane-ethane inversions in constraining *FF* sector is small, likely because of the strong spatial separation between *FF* and *non-FF* sources in the UK. Therefore, this two-gas inverse model may be even more important for quantifying sectoral emissions estimates in areas of the world where inventories are more uncertain and where there is greater

spatial and temporal overlap between sources.

**5    Conclusions**

We have presented a method of estimating sector-level emissions of a trace gas using using a Bayesian atmospheric inverse model, observations of a secondary co-emitted tracer and its emission ratio relative to the primary gas. We use methane and ethane, co-emitted from fossil fuel emissions sources, as an example to highlight the utility of this method. A critical

advancement of this work is in the inclusion of ethane:methane emission ratios as a variable parameter, with its own prior PDF and uncertainty. We show how this uncertainty is carried forward into posterior flux estimates to improve overall uncertainty characterisation. Through a synthetic data experiment and the UK case study, we show how errors can potentially be introduced





into posterior methane estimates if the ethane:methane emission ratio is assumed to be fixed but incorrect. Using a variable emission ratio and considering the uncertainty in this ratio mitigates these potential errors.

Using this model, we find average 2015-2019 UK methane emissions from fossil fuel sources of 0.27 (95% uncertainty interval 0.26-0.29) Tg y$^{-1}$ and from non-fossil fuel sources of 2.06 (1.99-2.15) Tg y$^{-1}$. The 95% uncertainty intervals of the UK total methane emission estimates made here are within the bounds of most previous estimates, (Ganesan et al., 2015; Zammit-Mangion et al., 2015; Lunt et al., 2016; Western et al., 2020) but fossil fuel emissions are 15% lower than when estimated using only methane observations and the spatial separation of emissions in the prior.

This inverse model is highly adaptable and could be used with other trace gases to constrain methane emissions from other target sources. For example, methane isotopologue observations could be used in place of ethane to estimate methane fluxes from a range of key sources. Recent developments in instrumentation, allowing for high frequency isotopologue observations are a promising target for future investigations of methane emissions with this method.

*Code and data availability.* Measurements of methane from the UK DECC network sites Tacolneston, Bilsdale, Ridge Hill and Heath-
field are available from https://catalogue.ceda.ac.uk (last access 24 August 2021). Measurements of methane from the Mace Head site are available from http://agage.mit.edu/data (last access 24 August 2021). Ethane observations from the Mace Head and Tacolneston sites are included in a supplementary data file. The NAME III v7.2 transport model is available from the UK Met Office under licence by contacting enquiries@metoffice.gov.uk. The meteorological data used to drive the transport model from the UK Met Office operational Numerical Weather Prediction (NWP) Unified Model (UM) are available from https://data.ceda.ac.uk/badc/ukmo-nwp (last access 24 August 2021). The
UK Greenhouse Gas (UKGHG) model is available from https://github.com/NERC-CEH/ukghg (Last access 24 August 2021). The EDGAR v5.0 methane inventory is available from https://edgar.jrc.ec.europa.eu/dataset_ghg60 (last access 24 August 2021). Data from the MOYA FAAM aircraft campaign is available from the Centre for Environmental Data Analysis (CEDA) archive, at https://catalogue.ceda.ac.uk/ (last access 24 August 2021). The code used to infer methane emissions using these data products, with an example month of data for testing, are available from https://doi.org/10.17605/OSF.IO/VH8ND (last access 24 August 2021). Any further data or code is available from the
corresponding author on request.

*Author contributions.* AR and AG led the method and code development, investigation and manuscript preparation. LW contributed to code development, methodology and manuscript editing. MR advised on the study. AM provided NAME footprints. PL provided the UKGHG methane flux model. AF, JF and PB collected and analysed the ethane:methane ratio validation data. DS, AW, TA, CR, KS, SO and DY made the measurements from the UK DECC network.

*Competing interests.* The authors declare that they have no conflicts of interest.



*Disclaimer.* TEXT

*Acknowledgements.* Alice Ramsden was funded by a NERC GW4+ Doctoral Training Partnership studentship from the Natural Environment Research Council (NERC). Anita Ganesan was funded by a NERC Independent Research Fellowship (NE/L010992/1). The project was supported by the NERC Detection and Attribution of Regional Emissions in the UK (DARE-UK) programme (NE/S004211/1). Measurements from Mace Head were funded by the Advanced Global Atmospheric Gases Experiment (NASA grant NNX16AC98G) and measurements from the UK DECC network by the UK Department of Business, Energy & Industrial Strategy through contract (1537/06/2018) to the University of Bristol. Since 2017, measurements at Heathfield have been maintained by the National Physical Laboratory mainly under funding from the National Measurement System. The MOYA FAAM North Sea flights were jointly funded by NERC and the United Nations Environment Programme: Climate and Clean Air Coalition (UNEP CCAC). We would like to give special thanks to the Airtask Ltd. pilots and engineers and all staff at FAAM Airborne Laboratory for their hard work in helping plan and execute the successful MOYA project flights. This work was carried out using the computational facilities of the Advanced Computing Research Centre at the University of Bristol. We would like to thank those that have contributed to the Bristol Atmospheric Chemistry Research Group's code repository.





**Appendix A: Numerical Atmospheric-dispersion Modelling Environment (NAME)**

NAME is a Lagrangian particle model (Jones et al., 2007) used to estimate the relationship between surface emissions and
atmospheric observations. The model simulated the transport of 20,000 inert gas particles from the measurement location each
hour, back in time for up to 30 days, and quantified their interaction with the surface and their exit locations/times from the study
domain. These hourly footprints were then averaged into four-hourly footprints, to match the averaging of the observations.
Meteorological data from the UK Met Office's Unified Model (Walters et al., 2014) and a nested UK-specific 1.5 km horizontal
resolution meteorological product were used to drive NAME at a one-hourly temporal resolution over the UK and at three-
hourly resolution over the rest of the domain. The output was stored at $0.23° \times 0.35°$ spatial resolution over a domain spanning
$-97.9°$ to $39.7°$E longitude, $10.7°$ to $79.3°$N latitude. This process was carried out for each observation made at each site, to
build up a field of emissions sensitivity for the whole domain.

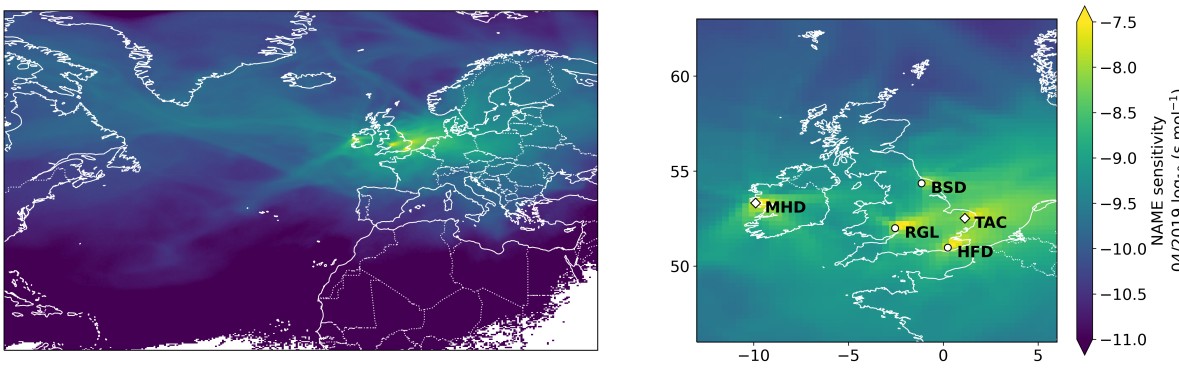

**Figure A1.** Monthly NAME sensitivities for May 2019 for the full NAME domain (left). Close up on the UK (right) showing locations of
the four tall tower observation sites and the coastal observation site used in this study: Mace Head (MHD) at $9.90°$W $53.33°$N, Tacolneston
(TAC) at $1.14°$E $53.52°$N, Bilsdale (BSD) at $1.15°$W $54.36°$N, Ridge Hill (RGL) at $2.54°$W $52.00°$N and Heathfield (HFD) at $0.23°$E
$50.98°$N. Areas with higher values have greater sensitivity to emissions from the surface. Sites with a diamond marker have both methane
and ethane observations. Site with a circular marker only have methane observations.



## Appendix B: FAAM ethane:methane emission ratios

Measurements of methane and ethane were made using an Aerodyne interband cascade laser (ICL), at a resolution of 1 Hz
415 (France et al., 2021). This data was used to identify time periods during flight C191 with concurrent enhancements in methane
and ethane. Two methods were used to derive ethane-methane ratio, 1. Regression analyses of ethane and methane mixing
ratios were performed for each enhancement. The slopes of these regressions were used to derive the ethane:methane ratio,
using a similar approach to Wilde et al. (2021). 2. A Gaussian curve was fit to each enhancement of methane and concurrent
ethane enhancement. The integral of each of these curves was then used to calculate the ethane-methane ratio of each methane
420 enhancement. Previous work in France et al. (2021) showed that consistency is expected between these two methodologies.





## Appendix C: A priori flux estimates

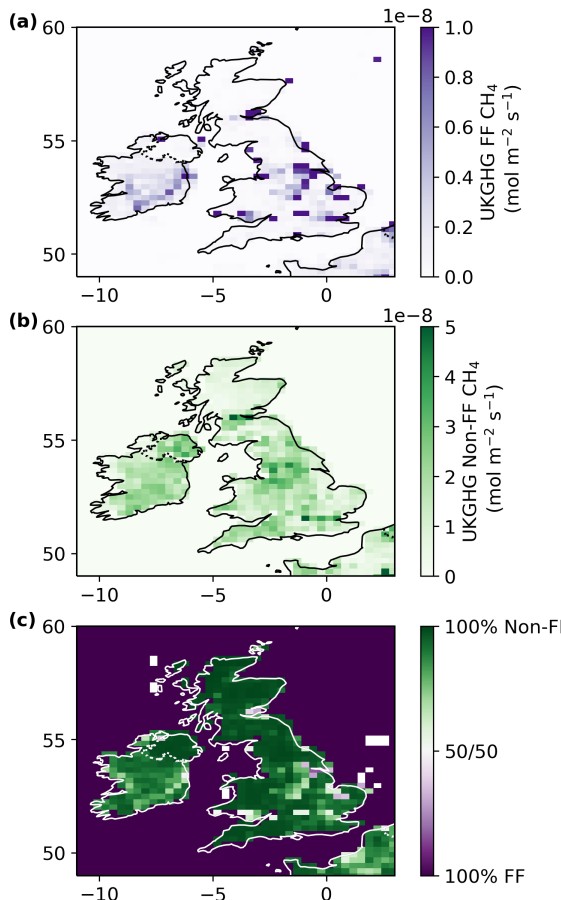

**Figure C1.** Monthly FF (a) and non-FF (b) methane emissions estimates from the UKGHG model for 2015. The percentage contribution of each source to the total emissions from each grid cell is given in (c), to illustrate the strong spatial separation of methane sources in the UK, and the dominance of emissions from the non-FF (primarily biogenic) sector.


## Appendix D: Posterior mole fraction comparisons

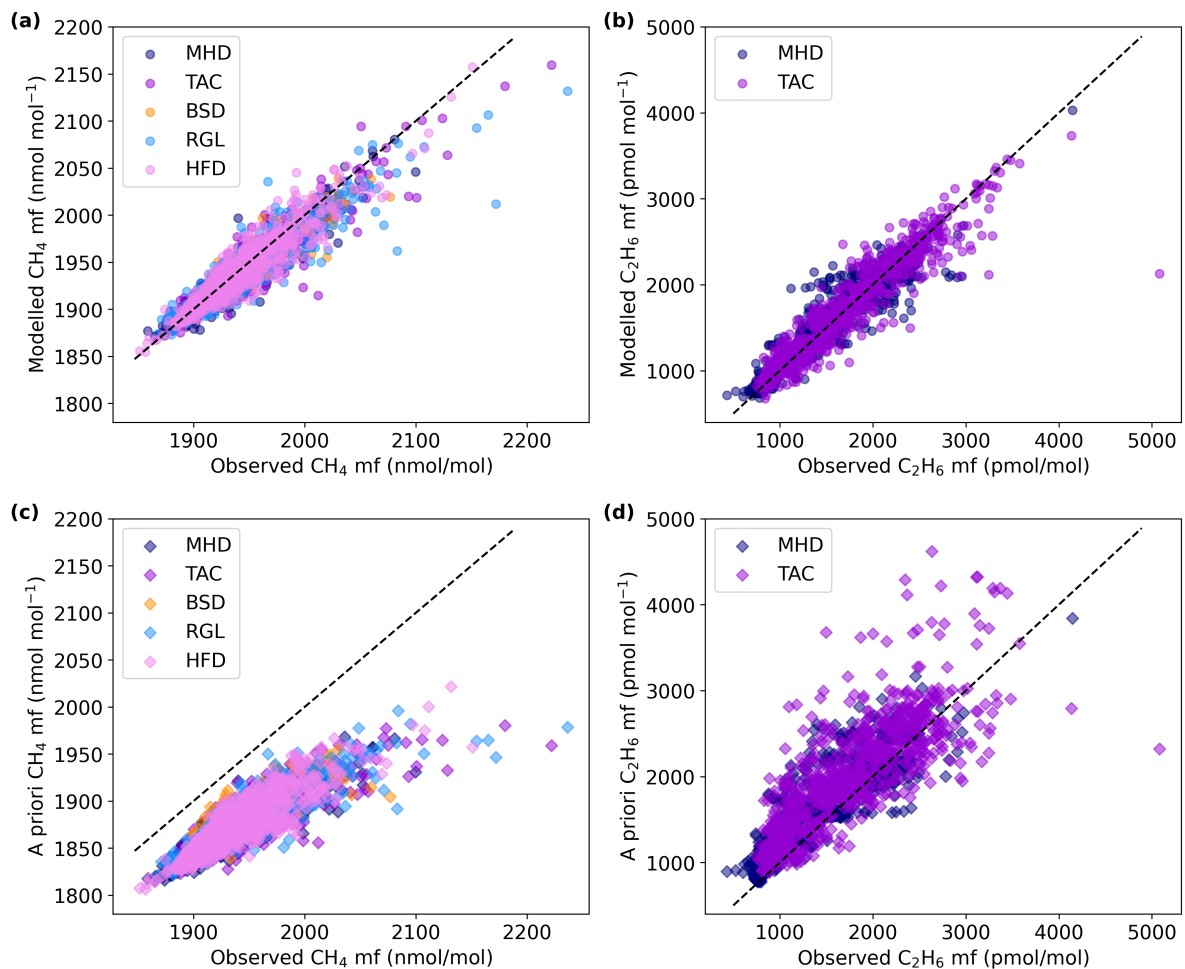

**Figure D1.** Scatter plot comparing mole fraction observations of methane (left) and ethane (right) as used in this study (2015-2019) to posterior mean modelled estimates of these values, made using the joint methane-ethane inversion (a,b). Comparison with a priori estimates of mole fraction concentrations made by combining the a priori flux maps with the transport model are also given (c,d) to show how the joint model improves the fit to observations. Points are colour-coded by site, with the locations of these sites given in Appendix Fig. A1

.



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
