# Peer review of "Quantifying fossil fuel methane emissions using observations of atmospheric ethane and an uncertain emission ratio"

_Atmospheric Chemistry and Physics, 2021_

## Author Comment (AC1)

**Reply to Referee Comments on acp-2021-734**

In the following document, referee comments are presented in black, followed by our replies in blue and changes to the manuscript in red.

**Reply to RC1**

**Overall comment:**

The paper by Alice E. Ramsden et al. describes a method for estimating fossil fuel methane emissions using airborne methane and ethane observations. The authors aim at reducing uncertainties in UK fossil fuel emission estimates by augmenting their inverse approach with an additional ethane-to-methane ratio parameter. They highlight the detrimental impact of an incorrect, fixed ethane-to-methane ratio on posterior emission estimates and the confidence level gain by the additional inversion parameter. In general, the paper is well written and the subject highly topical within the scope of ACP. I recommend publication after addressing some minor comments.

We thank the referee for their positive review of the manuscript and for their helpful comments and suggestions for changes required to improve our manuscript. We provide detailed responses to each of their comments below.

**Specific comments:**

p. 2, l. 21 Please indicate what "short atmospheric lifetimes" means for greenhouse gases. Methane does not have a short atmospheric lifetime.
We have updated this sentence as follows:
p. 2, l. 21: … Due to its lifetime of around a decade, and high impact…

p. 2, l. 46 Please indicate EDGAR version used in this study - v.4.3.2, v5? The reference indicates the use of EDGAR v5, however Chen et al., 2018 refer to EDGAR v4.2. Has the partitioning not improved since then?
We have added the model version and expanded this sentence to discuss how more recent versions have been improved:
p.2, l. 46-48: … Atmospheric Research v.4.2 (EDGAR, Team EDGAR 2021) were shown to be… production (Chen et al., 2018). Recent updates to this inventory (v.5.0 and 6.0) now include more detailed temporal and spatial profiles.

p. 3, l. 77 Did Yacovitch et al., 2017 and Lowry et al., 2020 really make use of aircraft observed plumes. To my understanding both of these studies are ground-based obs. Please remove "aircraft-observed" in that case.
Thank you for noticing this error. We have corrected this sentence to read:
p. 3, l. 77: …enhancements seen in individual plumes observed by ground-based vehicle-mounted instruments…

p. 5, l. 120-123 Atmospheric transport is influenced by the molar mass of the simulated species. Simulated plumes can therefore significantly differ. I wonder if the authors checked this statement for ethane having almost twice the molar mass compared to methane. Furthermore, ethane has a limited lifetime in the atmosphere. How does this influence the inversion given the timescales of the simulations?

We politely disagree with the reviewer that the molar mass will significantly affect atmospheric transport of these gases. The difference in gravitational settling will be negligible compared to the mixing that arises from turbulence, etc.

p. 8, l. 203-205 contains information on how we were able to assume equivalent transport footprints of methane and ethane. The timescale of transport between a source and receptor for these regional domains is around a couple of days, which is short compared to the summer lifetime of ethane (2 months). We now show in Appendix Fig B1 the average percentage difference between an example transport footprint for a gas with a lifetime of 2 months and a footprint for an inert substance. We have edited the text to point to this additional information:

p. 8, l. 207-208: …used the same transport footprints for both gases (see Fig. B1 for a comparison of a footprint for an inert gas and for a gas with a 2 month lifetime).

p. 5, l. 128 I wonder how exactly the model-measurement uncertainty has been estimated, as this is a very critical parameter for the inversion. On p. 9 l. 231 the uncertainty is described as the quadratic sum of measurements uncertainty and model uncertainty. Afterwards the measurement representation uncertainty is defined as 1 sigma from 1 minute of data. Is this also included in the quadratic sum? Model uncertainty is subsequently chosen with a uniform PDF between 10 and 50ppb (CH4) and 20-50ppb (C2H6) without justification. This reviewer wonders how transport model error is included. In general, the description of uncertainty lacks some details, as this is a very crucial part of the study when claiming to narrow down uncertainties, especially if the posterior emission estimates show a dependence on the prior (p. 10, l.259). In the same context, what is the justification for prior uncertainties?

First, there is a error in our sentence which should not say 'measurement *representation* uncertainty' but just 'measurement uncertainty'. We agree that model uncertainty is a crucial parameter in an inversion and is one of the main reasons we employ a hierarchical Bayesian inversion in which this parameter is estimated as part of the inversion.

We have rewritten the paragraph p. 9, l. 232-237 to add more detail:

p. 9, l. 235-241: Observational uncertainty includes both measurement and model uncertainty. Measurement uncertainty of 4 hourly data was calculated as the variability within the averaging period. Model uncertainty was included as a hyper-parameter, with one value per site per month solved for during the inversion. This model uncertainty was given a uniform PDF between 10 and 50 ppb for methane and between 20 and 50 ppb for ethane (these values were chosen based on results from previous work using these datasets (Ganesan et al. (2014)). Total uncertainty for each observation is calculated as the quadratic sum of the measurement uncertainty at each time point and the model uncertainty at each site. A full description of a similar use of model uncertainty in a hierarchical framework can be found in Ganesan et al. (2014).

p. 6, l. 155 Adding gaussian noise is a very humble approach to simulating instrument noise. I wonder if the authors also considered the influence of added systematic noise and possibly a bias?

The purpose of the synthetic-data test is to highlight the benefit of the method, rather than to try to capture the different unknown uncertainties in the true system. Thus, we keep the framework simple, only allowing this one parameter (emission ratio) to be unknown or incorrect and specifying other parameters (random instrument noise) exactly in the inversion. Thus, we respectfully do not think that the addition of systematic errors to be necessary for the synthetic-data proof-of-concept.

p. 6, l. 167 What kind of "basis functions" are referred to here?

We have reworded this sentence as follows and added Figure C1, which gives the arrangement of grid cells:

p. 6, l. 169-171: In all tests, the inversion solved for emissions as a scaling of the a priori emissions field, on a coarser grid than the native resolution of the transport model. The inversion estimated parameters for 49 regions over the UK, with the rest of the European domain split into four larger regions (see Fig. C1 for a representation of the inversion domain).

p. 14, Fig. 5 Is b) showing "Posterior std dev"? Looks like it is showing posterior means. Unfortunate, that there is so few ratio observations to compare to.

Thank you for pointing out this error in the colour bar title, we have corrected this to read 'Posterior mean R'.
* * *
**Spelling and grammar:**

p. 12, l. 290 is there a word missing? "[…], so there … little [...]"
We have corrected this sentence to read:
… so there is little difference in the spatial…

p. 13, l. 298 "[…] of the both [...]"
We have corrected this sentence to read:
… distribution of both the methane flux…

p. 16, l. 357 "[…] using using a […]"
We have corrected this sentence to read:
…trace gas using a Bayesian…

**Reply to RC2**
* * *
**Overall comment:**

I think the authors have put together a nice manuscript on ethane and methane emissions from the United Kingdom. I have a few minor suggestions and questions related to the manuscript, and I recommend the article for publication in ACP.

We thank the referee for their positive review of the manuscript and for their helpful comments and suggestions for changes to make to improve our manuscript. We provide detailed responses to each of their comments below.
* * *
**Overall suggestions:**

This study uses a spatially constant ethane:methane ratio, which likely makes sense for the United Kingdom. In many other countries, like the US, this ratio is spatially variable. In this manuscript, I think it would be useful to think about how the proposed inverse modelling framework could be applied in locations where the ratio is spatially heterogeneous. For example, I wasn't completely sure how one would use Eq. 5 for the case where the ethane:methane ratio is spatially variable.

We use a spatially constant ethane:methane ratio in the synthetic data tests to test the model with a simplified system. However, in the UK methane emissions case study (Sections 2.2 and 3.2-3.3) we used a spatially uniform emission ratio *prior* but solved for scaling factors of these ratios on the same inversion resolution as the emissions (these are shown in Fig. 5 and as discussed in p8., l. 222-227 and p.13, l. 299-308. In Eq. 5, the emission ratio parameter (**R**) is a matrix, of the same dimensions as the emissions scaling parameter (**x**), so the emission ratios can be included on any spatial scale. We have updated the following sections to more clearly explain how we used spatially varying emission ratios:

p. 1, l. 2: The ethane:methane emission ratio is incorporated as a spatially and temporally variable parameter in a …
p. 5, l. 120: …with emission ratios **R**, relative to…
p. 5, l. 128: … in the emission ratios, **R** would be imposed as fixed parameters
p. 16, l. 348-350: …also assume one possible range of emission ratios for all fuel types, rather than applying different ranges to, for example, coal or natural gas sources…
* * *
**Specific suggestions:**

Last paragraph on pg. 4: Which MCMC algorithm do you use (and/or build upon for the inverse model)?

We have updated this paragraph to include information about the MCMC algorithm used (Metropolis Hastings):

p. 4, l.106-107: This is an iterative method, based on the Metropolis Hastings algorithm, that randomly samples the…

Line 165: How does this case work if the spatial distinction of sources in the prior does not exist?
We show in Fig. 1 and p. 9, l. 247-250 that the model returns the prior separation of sources in the one-gas case when there is no prior difference in the spatial distribution of sources. As this is a synthetic data test, we chose an extreme example to illustrate how the two-gas model can separate sources in a situation that the one-gas is unable to.

Line 168: What is the unit on the "7 x 7" resolution listed here?
We agree that the 7x7 is confusing and have re-phrased to state that 49 regions over the UK were estimated in the inversion. We also added Figure C1 to illustrate this.
p. 6, l. 169-171: In all tests, the inversion solved for emissions as a scaling of the a priori emissions field, on a coarser grid than the native resolution of the transport model. The inversion estimated parameters for 49 regions over the UK, with the rest of the European domain split into four larger regions (see Fig. C1 for a representation of the inversion domain).

Lines 255 - 261: I'm guessing that the accuracy of the estimated emissions in case 2 depends on how you construct the prior pdf on the ethane:methane ratio. I.e., if the mean of this pdf were 50% higher or 50% lower than the true ethane:methane ratio, I assume that the estimated sector-specific methane emissions would also be relatively inaccurate. Line 296 explains that the inverse model converged on a Gaussian distribution for R only 20% of the time, and that makes me wonder even more about this point. I think an important advantage of this framework, however, is that the posterior uncertainty bounds should encompass the real value if the prior pdf is carefully constructed.

We agree that the emissions ratio has a large impact on the results and thus this approach is designed to consider uncertainty in this parameter. The prior PDF must be constructed based on expert knowledge and so we agree that the PDF must be based in reality. In this application, the emissions ratio PDF was constructed with a uniform distribution and this means that no value has higher probability than another within the bounds of the distribution. If the true value lies within the bounds, it will be captured in the statistics. The fact that the inversion converged on a Gaussian for 20% of the regions is mainly due to the fact that only 20% of the domain had enough sensitivity from the measurements (i.e. the footprints of the ethane measurements from two sites only strongly constrained 20% of the regions over the UK).

Lines 295-296: Do you think a range of 0.009 to 0.2 is a physically realistic range of values? (After reading the entire manuscript, I see that there is more discussion of this point in Sect. 3.4. You may want to refer the reader to this section for discussion of whether a range of 0.009 to 0.2 is realistic.).
It is realistic in that our prior uniform PDF was set with bounds of 0.0075 and 0.2 (as discussed p. 9, l. 238). These values were chosen to include the most common ratios found by bottom up estimates of European fossil fuel ethane:methane ratios from a range of studies and databases (Table 1)." We have added a reference to Sect. 3.4. as recommended:

p. 13, l. 300: …and 0.2 (we discuss the implications of these posterior ratio values in Sect. 3.4).

Appendices: You might consider moving the appendices to the supplement. With that said, I think it's a point of personal preference.
We have retained the current format of using appendices, as we like the ease to which the appendix material can be accessed within the same document. The figures within the appendix are important to understanding the manuscript but would disrupt the flow of the text if included elsewhere.

---

## Editor Decision (ED1)

Editor corrections on ACP-2021-734

Quantifying fossil fuel methane emissions using observations of atmospheric ethane and an uncertain emission ratio by Ramsden et al.

Technical corrections:
* * *
P1, L8: Add "the" so that it reads "......to the Climate Change (DECC) network."
P2, L24: upwards trend -> upward trend
P3, L59-60: tracer gases -> trace gases
P3, L66: emissions estimates -> emission estimates
P3, L80: emissions profiles -> emission profiles
P3, L83: rather plural? Thus "Most of the previous works using observations of methane have used....."
P4, L105: is used produce -> is used to produce
P5, L139: emissions field -> emission field
P6, L150: add "the" -> the Climate Change......
P6, L166: emission ratio -> emission ratios
P6, L169: emissions field -> emission field
P6, L172: for at -> at
P6, L173: emissions field -> emission field
P7, L175: emissions field -> emission field
P7, L197: add "the ones" so that it reads "....as the ones described in......"
P8, L206: one closing parentheses obsolete
P8, L218: skip "Appendix" and just write "in Fig. E1" or write "in the Appendix in Fig. E1"
P8, L225: add "is" -> The inversion is then.....
P8, L225: Sentence not clear. a scaling factor derived form the a priori?
P8, L230: rather "at" or "to" negative emissions than "on" negative emissions
P8, L230: given a mean -> given with a mean
P9, L237: rather "for" or "during" than "for during"
P9, L237: given a uniform -> given as a uniform
P9, L239: One closing parentheses obsolete
P10, L253: a value 50% ...-> a value "of" (or "which is") 50% lower......
P10, L256: Don't start the sentence with a number -> Add "The", thus "The 95%......"
P10, L272: finds -> shows
P10, L272: 15% less to total -> 15% less to the total
P10, L273: skip "in" and just write "the methane-only inversion"
P10, L274: move "of the posterior FF flux" at the end of the sentence
P12, L287: write "in Fig F1" or "in the Appendix in Fig. F1"
P12, Fig 3 caption: Add "Left" and "Right"
P12, L291: Add "the", so that it reads "converge the methane...."
P12, L291: What do you mean with model measurement uncertainty? It should be either model uncertainty or measurement uncertainty or do you mean a modelled measurement uncertainty?
P13, L293: emissions scaling factors -> emission scaling factors
P13, Figure 4 caption: emissions scaling factors -> emission scaling factors
P13, L300: in 3.4 -> in Sect. 3.4
P14, Figure 5 caption: second and third line change "ratio" to "ratios"
P14, Figure 5 caption: from Royal Holloway ...... -> from the Royal Holloway.....
P15, L322: remove "to have"
P19, L410: Either write "The model simulates"
P19, Figure A1 caption: Site -> sites
P20, Figure B1 caption: Add "Top:" and change "Bottom shows" to "Bottom: The percentage ......"

P22, L423: ratio -> ratios

P24, Figure F1 caption: remove "Appendix" and just write "in Fig A1".

P28, L570: check author list, something went wrong here.

P29, L589 and 596: same here as for P28, L570.

---

## Author Response (AR2)

Dear Farahnaz Khosrawi,

Thank you for the acceptance of our manuscript *Quantifying fossil fuel methane emissions using observations of atmospheric ethane and an uncertain emission ratio* for publication. Below we detail the technical corrections made to the manuscript as requested.

Kind regards,

Alice Ramsden
* * *
We have made the following changes as requested:

P2, L24: upwards trend -> upward trend
P3, L59-60: tracer gases -> trace gases
P3, L66: emissions estimates -> emission estimates
P3, L80: emissions profiles -> emission profiles
P3, L83: rather plural? Thus "Most of the previous works using observations of methane have used....."
P4, L105: is used produce -> is used to produce
P5, L139: emissions field -> emission field
P6, L169: emissions field -> emission field
P6, L172: for at -> at
P6, L173: emissions field -> emission field
P7, L175: emissions field -> emission field
P7, L197: add "the ones" so that it reads "....as the ones described in......"
P8, L218: skip "Appendix" and just write "in Fig. E1" or write "in the Appendix in Fig. E1"
P8, L225: Sentence not clear. a scaling factor derived form the a priori?
P8, L230: rather "at" or "to" negative emissions than "on" negative emissions
P9, L237: rather "for" or "during" than "for during"
P10, L253: a value 50% ...-> a value "of" (or "which is") 50% lower......
P10, L256: Don't start the sentence with a number -> Add "The", thus "The 95%......"
P10, L272: 15% less to total -> 15% less to the total
P10, L274: move "of the posterior FF flux" at the end of the sentence
P12, L287: write "in Fig F1" or "in the Appendix in Fig. F1"
P12, Fig 3 caption: Add "Left" and "Right"
P12, L291: Add "the", so that it reads "converge the methane...."
P12, L291: What do you mean with model measurement uncertainty? It should be either model uncertainty or measurement uncertainty or do you mean a modelled measurement uncertainty? We have corrected this to 'model uncertainty'.
P13, L293: emissions scaling factors -> emission scaling factors
P13, Figure 4 caption: emissions scaling factors -> emission scaling factors
P13, L300: in 3.4 -> in Sect. 3.4
P14, Figure 5 caption: from Royal Holloway ...... -> from the Royal Holloway.....
P15, L322: remove "to have"

P20, Figure B1 caption: Add "Top:" and change "Bottom shows" to "Bottom: The percentage ......"
P22, L423: ratio -> ratios
P24, Figure F1 caption: remove "Appendix" and just write "in Fig A1".
P28, L570: check author list, something went wrong here.
P29, L589 and 596: same here as for P28, L570.
* * *
We have not made these changes for the following reasons:

P8, L225: add "is" -> The inversion is then.....
P8, L230: given a mean -> given with a mean
P9, L237: given a uniform -> given as a uniform
P10, L272: finds -> shows
P10, L273: skip "in" and just write "the methane-only inversion"
P19, L410: Either write "The model simulates"
P19, Figure A1 caption: Site -> sites
These grammatical changes would adjust the meaning of the text.

P1, L8: Add "the" so that it reads "......to the Climate Change (DECC) network."
P6, L150: add "the" -> the Climate Change......
We have not made these changes, as the official acronym for the DECC network is 'Deriving Emissions linked to Climate Change'

P6, L166: emission ratio -> emission ratios
P14, Figure 5 caption: second and third line change "ratio" to "ratios"
A single emission ratio was used in this test.

P8, L206: one closing parentheses obsolete
P9, L239: One closing parentheses obsolete
The second closing parenthesis is paired with the opening parenthesis on the previous line.

[revised manuscript text omitted]